# Effect of Smartphone Use on Sleep in Undergraduate Medical Students: A Cross-Sectional Study

**DOI:** 10.3390/healthcare11212891

**Published:** 2023-11-02

**Authors:** Ashish Goel, Arsalan Moinuddin, Rajesh Tiwari, Yashendra Sethi, Mohammed K. Suhail, Aditi Mohan, Nirja Kaka, Parth Sarthi, Ravi Dutt, Sheikh F. Ahmad, Sabry M. Attia, Talha Bin Emran, Hitesh Chopra, Nigel H. Greig

**Affiliations:** 1Graphic Era Institute of Medical Sciences, Dehradun 248008, Uttarakhand, India; dr.ashishgoel2012@gmail.com (A.G.); dr.ravidutt@gmail.com (R.D.); 2School of Sport and Exercise, University of Gloucestershire, Gloucester GL50 2RH, UK; 3Gautam Buddha Chikitsa Mahavidyalaya, Dehradun 248007, Uttarakhand, India; yssrajeshtewari@yahoo.in; 4Government Doon Medical College, Dehradun 248001, Uttarakhand, India; yash@pearresearch.com; 5PearResearch, Dehradun 248001, Uttarakhand, India; nirja@pearresearch.com; 6Moray House, University of Edinburgh, Edinburgh EH8 8AQ, UK; 7Veer Chandra Singh Garhwali Government Institute of Medical Science and Research, Srinagar 246174, Uttarakhand, India; aditimohan2706@gmail.com; 8GMERS Medical College, Himmatnagar 382012, Gujarat, India; 9Rajkiya Medical College, Jalaun 395001, Uttar Pradesh, India; drparthsarthi@gmail.com; 10Department of Pharmacology and Toxicology, College of Pharmacy, King Saud University, Riyadh 11451, Saudi Arabia; fashaikh@ksu.edu.sa (S.F.A.); attiasm@ksu.edu.sa (S.M.A.); 11Department of Pathology and Laboratory Medicine, Warren Alpert Medical School & Legorreta Cancer Center, Brown University, Providence, RI 02912, USA; 12Department of Pharmacy, Faculty of Allied Health Sciences, Daffodil International University, Dhaka 1207, Bangladesh; 13Department of Biosciences, Saveetha School of Engineering, Saveetha Institute of Medical and Technical Sciences, Chennai 602105, Tamil Nadu, India; hodbiotech.sse@saveetha.com; 14Drug Design & Development Section, Translational Gerontology Branch, Intramural Research Program, National Institute on Aging, National Institutes of Health, Baltimore, MD 21224, USA; greign@grc.nia.nih.gov

**Keywords:** mobile phone, sleep disturbances, sleep, medical students

## Abstract

Smartphone use, particularly at night, has been shown to provoke various circadian sleep–wake rhythm disorders such as insomnia and excessive daytime tiredness. This relationship has been mainly scrutinized among patient groups with higher rates of smartphone usage, particularly adolescents and children. However, it remains obscure how smartphone usage impacts sleep parameters in adults, especially undergraduate college students. This study sought to (1) investigate the association between smartphone use (actual screen time) and four sleep parameters: Pittsburgh sleep quality score (PSQI), self-reported screen time, bedtime, and rise time; (2) compare the seven PSQI components between good and poor sleep quality subjects. In total, 264 undergraduate medical students (aged 17 to 25 years) were recruited from the Government Doon Medical College, Dehradun, India. All participants completed a sleep questionnaire, which was electronically shared via a WhatsApp invitation link. Hierarchical and multinomial regression analyses were performed in relation to (1) and (2). The average PSQI score was 5.03 ± 0.86, with approximately one in two respondents (48.3%) having a poor sleep index. Smartphone use significantly predicted respondents’ PSQI score (β = 0.142, *p* = 0.040, R^2^ = 0.027), perceived screen time (β = 0.113, *p* = 0.043, R^2^ = 343), bedtime (β = 0.106, *p* = 0.042, R^2^ = 045), and rise time (β = 0.174, *p* = 0.015, R^2^ = 0.028). When comparing poor-quality sleep (PSQI ≥ 5) to good-quality sleep (PSQI < 5), with good-quality sleep as the reference, except sleep efficiency and sleep medications (*p* > 0.05), five PSQI components declined significantly: subjective sleep quality (β = −0.096, *p* < 0.001); sleep latency (β = −0.034, *p* < 0.001); sleep duration (β = −0.038, *p* < 0.001); sleep disturbances (β = 1.234, *p* < 0.001); and sleep dysfunction (β = −0.077, *p* < 0.001). Consequently, public health policymakers should take this evidence into account when developing guidelines around smartphone use—i.e., the when, where, and how much smartphone use—to promote improved sleep behaviour and reduce the rate of sleep–wake rhythm disorders.

## 1. Introduction

Contrary to the largely stationary internet world of the early 2000s, the vast majority of the Indian population today (77%), like numerous industrialized and developing countries, has become increasingly connected to the world of digital information via smartphones, and this trend continues at a staggering pace [1]. A smartphone is a mobile phone that encompasses a myriad of computer functions, including a touch screen interface, internet access, and an operating system to download applications [2]. In addition to their use for communication, smartphones can amass and process a plethora of information compared to an ordinary cell phone, e.g., games, social networks, videos, multimedia, and navigation. As such, the smartphone praxis routine involves extended periods of usage, surfing the internet, using social media, playing games, etc., plenty of which have emerged as potential health risk factors [3,4,5]. Further to these contingency factors, hassle-free access to the internet also contributes to widespread smartphone use in India, as it does in many other countries.

Per the ‘Statista’ survey (2021) of smartphone ownership, the number of Indians using smartphones has increased from 304 million (22%) in 2016 to 760 million (54%) in 2021 [6] Whereas smartphone ownership in India still exhibits wider variation based on age, household income, and educational status, desktop computer or laptop ownership is still limited in India (20%). This appears to be in stark contrast to profound smartphone use in India [6]. Notably, a steady decline in smartphone usage is prevalent amongst those who use it as their primary source of internet access in their homes. Only 10% of Indians are “smartphone-only” users, meaning that they own a smartphone, but do not have a traditional home internet service [6,7]. However, as of late, a steep rise in smartphone use for online access has been observed amongst younger adults (18–29 years), with as many as 22% admitting to checking their phones every few minutes [8,9]. Further, around 65% of adult smartphone users sleep with their phones turned on right next to them, while college students find it challenging to put aside their smartphones, even during sleep [10]. Notably, nearly all younger adults (90%) prefer to be close to their smartphones when they sleep at night; not surprisingly, most feel positive whenever their phone is within their vicinity [6,10].

Insomnia, or poor sleep quality, has emerged as a related public health issue, particularly in technologically advanced societies. In the past few decades, exposure to artificial light at night (ALAN) has witnessed an exponential rise, with an annual increase of 3 to 6% and more than 2% growth in intensity and radiance [11]. ALAN has been shown to positively influence alertness, physical activity, and cognitive performance, but this exposure in the evening and at night can suppress and delay normal melatonin secretion. Blue light exposure has shown to have various positive effects in some studies, but the effects on sleep are mostly negative and cannot be ignored. Typically, the effect of ALAN (magnitude and direction) on different phases of the circadian rhythm is delineated by the ‘phase response curve (PRC)’, which highlights that blue light (ALAN) emitted from smartphones and other electronic devices (1) causes early biological night delays (phase-shift) in the circadian rhythm and melatonin secretion, and (2) is linked with insomnia, other sleep ailments, fatigue, and mental disorders [12,13,14]. Furthermore, both a lack of sleep and a poor sleep pattern negatively impact the physiological and mental well-being of adolescents. This is associated with a craving for high-calorie foods, a greater likelihood of alcohol abuse, self-harm, suicidal tendencies, excessive internet use, and smoking and drinking, particularly at night [15,16]. Limited studies thus far have examined the independent predicted effects of smartphone overuse on sleep parameters, specifically amongst the collegiate community, and concurrently controlling for confounders (socioeconomic status, literacy level, and personality traits). As such, to evaluate a potential link between sleep and mobile phone use, this study sought to (1) investigate the association between smartphone use (actual screen time) and quality of sleep as assessed by the Pittsburgh sleep quality score (PSQI); (2) to compare the seven PSQI components between good and poor sleep quality subjects in a medical college in north India, as an example of a young adult, educated population within a newly industrialized nation and a rapidly developing country.

## 2. Materials and Methods

### 2.1. Recruitment

A total of 264 undergraduate medical students (aged 17 to 25 years) were recruited from the Government Doon Medical College, Dehradun, India (between 15 March 2021 and 25 March 2021). A purposive sampling technique was used, and sample size was calculated using the G*power (version 3.1) software; the sample size for this study was calculated as 210–225 participants in order to reach the desired statistical power: β > 0.80; α < 0.05; and effect size ‘0.036’. However, 264 participants were recruited to account for attrition and unknown sources of error. Inclusion criteria: both English- and Hindi-speaking students aged 17 to 25 years of both genders, attending MBBS 1st and 2nd professionals, batches 2018 and 2019, respectively. Exclusion criteria: substance use, bipolar disorder, severe conduct disorder, and autism spectrum disorders.

### 2.2. Data Collection

We designed a survey questionnaire with structured questions on sleep parameters and mobile phone use. The questionnaire was accessed via a WhatsApp invitation link (in an electronic form) to undergraduate students of MBBS 1st and 2nd professionals, batches 2018 and 2019, respectively, from Government Doon Medical College, Dehradun, India. Out of approximately 350 students attending the school, a total of 264 responded by the collection date of 30 May 2021.

### 2.3. Ethics, Approval, and Consent to Participate

All participants provided their informed consent to participate in the research with full knowledge of the associated benefits and risks. Consent was provided by ticking ‘agree’, indicating the participants’ agreement to provide their information. This study was conducted within the regulations codified by the Declaration of Helsinki. It was further approved by the Institutional Review Board (IRB) of Government Doon Medical College, Dehradun, Uttarakhand, India (IRB#IEC/GDMC/2020/75).

### 2.4. Sleep Questionnaire

The questionnaire had two parts: (1) demographic data of age, gender, MBBS professional, socioeconomic status (SES), family structure, personal habits (such as smoking or alcohol consumption), use of a smartphone (hour/day), and the perception of smartphone overuse in terms of health hazards; (2) sleep parameters measured through the Pittsburgh Sleep Quality Index (PSQI) (overall reliability coefficient (Cronbach’s alpha of 0.736). This included questions about the participants’ usual sleep habits (days and nights) during the past month. The PSQI primarily determines sleep quality using seven sleep assessment indicators: (1) subjective sleep quality, (2) sleep latent period, (3) sleep time, (4) sleep efficiency, (5) sleep difficulties, (6) daytime functional impairment, and (7) use of sleeping pills. Scores on each item range from 0 to 3 points, and hence total scores range from 0 to 21 points, with a total PSQI score of <5 indicating good sleep quality, whereas a score of ≥5 indicates poor sleep quality.

### 2.5. Data Analysis

The statistical package SPSS for Windows (Version 27.0, Chicago, IL, USA) was used for all analyses. The comparative analysis was conducted on a sample size of 214 between the 2018 and 2019 batches, combined for both males and females, first qualitatively assessed using demographic analysis and normality assumptions (Kolmogorov–Smirnov and Shapiro–Wilk tests). The ‘hierarchical multiple regression’ was used to predict the extent to which smart phone use affected the sleep variables. The possible confounders—age, gender, parental structure, and socioeconomic status—were entered into the first step of a regression analysis, followed by the perceived and actual screen time in Step 2 and Step 3, respectively. To further examine the link between individual PSQI components and smartphone use, we utilized multinomial regression analysis.

## 3. Results

### 3.1. Overview

We studied the effect of smartphone overuse on the sleep quality of 264 medical students. The results were analysed using hierarchical and multimodal regression analysis. Collectively, we found excessive smartphone usage directly affected sleep quality. We further determined a possible association between smartphone use and respondents’ age, parental structure, socioeconomic status, actual screen time, and perceived screen time. Of interest, the duration of sleep was severely affected by family structure, particularly in respondents living with a single parent.

### 3.2. Baseline Characteristics

The 264 students (138 (52.3%) females) were from the 2018 and 2019 batches of Government Doon Medical College (Dehradun) aged between 17 and 25 years (21 ± 4 years). In total, 90.5% of respondents were from middle SES, 6% from lower SES, and 3.5% from high SES. A total of 190 (71.9%) respondents had both parents, whereas 11 (4%) lived with a single parent. All respondents owned a smartphone and had associated internet access; approximately 90% reported that they take their smartphone to bed with them prior to sleeping.

### 3.3. Smart Phone Use (Actual Screen Time) and Sleep Parameters

The average PSQI score was 5.03 ± 0.86; approximately one in two respondents (48.3%) had a poor sleep index, as defined by exceeding the cut-off point of five on the PSQI scale. There were significant positive relationships between smartphone use (actual screen time) and PSQI score (r = 0.182, *p* = 0.003), bedtime (r = 0.140, *p* = 0.023), and rise time (r = 0.154, *p* < 0.012). As presented in Table 1, compared to step (1) of the hierarchical regression analysis, the additional perceived screen time in step (2) and actual screen time in step (3) significantly increased variance in each model. In step (3), smartphone use (actual screen time) before going to bed significantly predicted respondents’ PSQI score (β = 0.142, *p* = 0.040, R^2^ = 0.027), perceived screen time (β = 0.113, *p* = 0.043, R^2^ = 343), bedtime (β = 0.106, *p* = 0.042, R^2^ = 045), and rise time (β = 0.174, *p* = 0.015, R^2^ = 0.028) (Table 1).

### 3.4. Smart Phone Use (Actual Screen Time) and Individual PSQI Components

Using hierarchical regression analysis, Table 2 presents the individual sleep components’ prediction based on sleep quality score (PSQI). Since limited respondents fit the highest category of each component, we used multinomial regression analyses to analyse individual components ranging from 0 to 2, thereby comparing poor-quality sleep (PSQI ≥ 5) to good-quality sleep (PSQI < 5), with good-quality sleep as the reference. (i) Subjective sleep quality: Although ubiquitously used, usually it is poorly defined as short sleep, characterized by tiredness on waking and throughout the day. Compared to those who had good-quality sleep (PSQI < 5), participants with poor-quality sleep (PSQI ≥ 5) had their subjective sleep quality score decreased by 0.096 for every 1 min increase in smartphone actual screen time (β = −0.096, *p* < 0.001). (ii) Sleep latency: The time taken to fall asleep after going to bed. Participants with poor-quality sleep (PSQI ≥ 5) had their sleep score decreased by 0.034 for every 1 min increase in actual screen time (β = −0.034, *p* < 0.001) compared to participants who had good-quality sleep (PSQI < 5). (iii) Sleep duration: The total quantity of sleep obtained, either nocturnal or acquired, across the 24 h period. The sleep score of participants with poor-quality sleep (PSQI ≥ 5) decreased by 0.0348 for every 1 min increase in actual screen time (β = −0.038, *p* < 0.001) compared to participants who had good-quality sleep (PSQI < 5). (iv) Sleep efficiency is the percent proportion of time asleep in relation to time spent in bed (approximately ≥ 85% is considered normal). The participants with poor-quality sleep (PSQI ≥ 5) had their subjective sleep quality score reduced by 0.878 for every 1 min increase in actual screen time (β = −0.878, *p* < 0.001) compared to participants who had good-quality sleep (PSQI < 5). (v) Sleep disturbances encompass disorders and dysfunctions associated with sleep, its stages, and partial arousals. Participants who had poor-quality sleep (PSQI ≥ 5) had their sleep disturbance score increased by 1.234 for every 1 min increase in actual screen time (β = 1.234, *p* < 0.001) in contrast to participants who had good-quality sleep (PSQI < 5). (vi) Use of sleep medication: Hypnotic medicines that induce sleep. Participants with poor-quality sleep (PSQI ≥ 5) had their sleep medication use score increased by 0.191 for every 1 min increase in actual screen time (β = 0.191, *p* = 0.07) compared to participants who had good-quality sleep (PSQI < 5). (vii) Daytime dysfunction: Inability to maintain wakefulness during waking hours. Compared to participants who had good-quality sleep (PSQI < 5), those who had poor-quality sleep (PSQI ≥ 5) had a decline in their daytime dysfunction score of 0.077 for every 1 min increase in actual screen time (β = −0.077, *p* < 0.001).

## 4. Discussion

The current study sought to (1) investigate the association between smartphone use (actual screen time) and four sleep parameters: Pittsburgh sleep quality score (PSQI), self-reported screen time, bedtime, and rise time, and (2) compare the PSQI seven components between good and poor sleep quality subjects in a medical college in North India. In line with previous studies, our data reported two key observations. First, smartphone use before going to bed was negatively associated with sleep outcomes. Second, we found significant interactions between smartphone use and rise time and most domains of the PSQI components. Our results are in close accord with previously published studies in children [17], adolescents [18], and adults [19], describing worse sleep outcomes with excessive smartphone usage, particularly at night. In contrast, Park et al. [20] reported smartphone use associated with later bedtimes, but this was unrelated to sleep disturbance.

Almost all living organisms need sleep to perform at their optimal capacity, with many vital processes, such as memory consolidation, body healing, metabolic regulation, etc., occurring during sleep. A lack of sleep or a poor sleep pattern can alter healthy digestion, body temperature, hormone release, etc., and ultimately hasten the development of chronic ailments such as diabetes, obesity, depression, and other sleep disorders [21,22,23,24]. The sleep–wake cycle in human beings is largely regulated by the circadian rhythm: a 24 h synchronized day/night cycle that is predominantly orchestrated by the suprachiasmatic nucleus (SCN) of the hypothalamus in the brain. Light is a non-visual stimulus detected first by the intrinsically photosensitive retinal ganglion cells (ipRGCs) of the retina, which contain melanopsin, and is then transmitted directly to the SCN in the brain. This cycle uses external cues such as light and melatonin, a hormone released in darkness that initiates the sleep versus awake cycle [12].

Smartphones utilize light-emitting diode (LED) backlights to enhance daytime brightness and contrast. The specifics of the light source used as the backlight will define the properties of the light emitted from a given smartphone [25]. Polychromatic “white” LEDs are routinely generated by fusing a blue LED with a yellow phosphor. This creates a light that appears white but possesses a spectral distribution that peaks in the blue segment of the electromagnetic spectrum [17]. The emitted blue light commonly has a wavelength close to the peak sensitivity for non-visual circadian photoreception. Chronic exposure to such blue light, even at low intensities, is considered to be above the predicted threshold for melatonin suppression. This, consequentially, likely disrupts the circadian rhythm, thereby potentially influencing a wide array of health outcomes: alertness, cognition, sleep, and activity levels (Figure 1) [12,13,14].

Several factors possibly explain our investigation. For instance, smartphone use before going to bed potentially disturbs sleep due to alterations in cognitive, emotional, or physiological mechanisms, likely due to light emission from the smartphone’s screen [18,19]. The regulation of the cycle of sleep/wakefulness relies on the circadian clock for which the SCN within the anterior hypothalamus is the master regulator [20]. SCN neurons possess a near 24 h rhythm of electrical activity, which is higher during the day and lower at night, even in the absence of environmental stimuli. This derives from the rhythmic expression of a core group of clock genes that are synchronized via transcriptional, translational, and post-translational regulatory mechanisms through multiple negative feedback loops [21]. The endogenous rhythm of the SNC is transmitted across the body by efferent neural and humoral signals, with pineal gland melatonin generation and secretion being particularly important among these. Melatonin regulation by the SCN involves the coordination of multiple neural pathways as well as feedback systems [12,13,14,15,22]. Under routine circumstances, SCN activity is fine-tuned and reset on a daily basis by light input via the retina, mediated by the retino-hypothalamic tract, during the day and by melatonin during darkness. The retino-hypothalamic tract comprises melanopsin-containing ipRGCs that, in particular, respond to light in the short-wave (blue) spectrum (principally 460 to 480 nm) [23,24], and whose input is relayed to the SNC via the optic nerve. The subsequent release of glutamate and pituitary adenylate cyclase-activating polypeptide regulates SNC activity that, in turn, controls pineal gland melatonin secretion via inhibitory projection to the paraventricular nucleus of the hypothalamus, whose signal is ultimately projected through the sympathetic system [21,26]. The key roles of melatonin are to provide feedback to the SNC and to provide information regarding the daily light–darkness cycle to target body structures where it impacts the sleep–wake cycle, and a plethora of other physiological factors, namely core temperature, heart rate, immune function, fat oxidation, mood, and cognitive functions. Hence, the suppression of melatonin secretion by smartphone blue light can inadvertently modify a multitude of physiological functions that, chronically, support the development of a broad variety of ailments [27,28,29,30,31]. Therefore, despite the debatable positive impacts of blue light exposure on alertness, physical activity, and cognitive performance, the effects on sleep quality are majorly negative and cannot be ignored [12,13,14,27].

Also, there is increasing evidence of electromagnetic field emission from smartphones varying sleep electroencephalograms (EEGs). Smartphone electromagnetic field exposure for long hours before going to bed has been reported to not only rearrange both circadian and melatonin rhythm by influencing brain activity, particularly of the pineal gland [32], but also modify sleep architecture and EEG slow-wave activity [33], with blue having significantly different effects to green or no light. In another study, prolonged exposure to smartphone emissions was reported to impact the melatonin onset time [33]. Blue light exposure during the day, however, is considered critical for circadian entrainment and overall well-being [23], with reported beneficial effects on alertness, mood, and productivity [34,35]. In this light, blue light daytime exposure from smartphones would not be expected to adversely impact circadian rhythm. Hence, the impact of blue light on the sleep–wake cycle and circadian entrainment is contingent on when exposure happens, in line with the phase response curve [17]. Ordinarily, early morning light exposure moves (i.e., phase advances) the circadian rhythm to earlier, while nighttime exposure to light shifts (i.e., phase delays) the circadian rhythm to later. Midday exposure has comparatively little impact [17,36]. In contrast to the debate regarding the daytime impact of blue light on the eye, there is a broad agreement that chronic and/or repeated nighttime blue light exposure has negative consequences on circadian health [17], which our study aligns with.

## 5. Limitations

Several limitations should be considered when interpreting our results. First, our cross-sectional design was susceptible to biases such as residual confounding and reverse causality. Second, our sample group was constrained to first- and second-year MBBS students only. Since all students were hostelers, the study cannot be generalized to those students living at home/day scholars, which can add potential unknown confounders. Questionnaire-based responses made our study prone to both recall and memory biases. Unfortunately, due to the limited data, we were unable to perform a subgroup analysis for sex differences.

## 6. Conclusions and Future Directions

The results of our study indicate that self-reported smartphone use before going to bed is highly prevalent amongst undergraduate medical students and that it is closely linked with use before going to bed. Clinicians, parents, and educators should be aware of the pervasiveness of problematic smartphone use and be prepared to consider the potential wide-reaching impact of smartphones on sleep. As such, public health policymakers should take this evidence into account when developing guidelines around smartphone use—that is, the when, where, and how much smartphone use—to promote improved sleep behaviour and a reduction in the rate of sleep–wake rhythm disorders.

## Figures and Tables

**Figure 1 healthcare-11-02891-f001:**
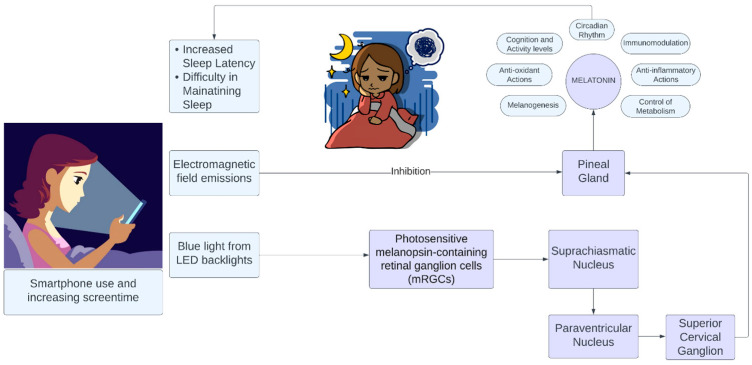
Effect of smartphone on sleep figure attribution. Made using Lucidchart (Lucid.io, accessed on 15 August 2023).

**Table 1 healthcare-11-02891-t001:** Hierarchical regression analyses: mobile phone use predicting sleep quality score (PSQI), self-reported screen time, actual screen time, sleep duration, bedtime, and rise time (N = 264); * *p* < 0.05.

Variables		PSQI		Perceived Screen Time		Bedtime		Rise Time	
		β	SE	β	SE	β	SE	β	SE
Step 1	Gender	0.080	0.338	0.052	0.234	0.061	0.268	−0.034	0.177
	Age	0.013	0.398	0.033	0.276	−0.003	0.316	0.108	0.208
	Parental St	0.128	0.663	0.063	0.459	0.072	0.526	−0.079	0.347
	SES	0.027	0.400	0.144	0.277	−0.034	0.318	0.023	0.209
	R^2^	0.017		0.012		0.006		0.003	
Step 2	Gender	0.073	0.337	0.002	0.206	0.054	0.267	−0.042	0.175
	Age	0.009	0.397	−0.006	0.242	−0.007	0.315	0.103	0.206
	Parental St	0.120	0.661	−0.006	0.406	0.064	0.524	−0.089	0.344
	SES	0.010 *	0.402	0.090	0.244	−0.052	0.319	0.001	0.209
	Perceived screen time	0.105	0.090	0.492	0.054	0.125	0.071	0.153	0.047
	R^2^	0.134		0.244		0.015		0.022	
Step 3	Gender	0.062	0.336	−0.266	0.311	0.046	0.268	−0.051	0.175
	Age	0.029	0.395	−0.096	0.132	−0.014	0.315	0.094	0.206
	Parental St	0.127	0.662	−0.059	0.328	0.053	0.527	−0.103	0.346
	SES	0.004	0.400	−0.716	0.632	−0.057	0.319	0.004	0.208
	Perceived screen time	0.048	0.102	0.090	0.663	0.074	0.082	0.054	0.076
	Actual screen time	0.142	0.101	0.113	0.070	0.106	0.080	0.174	0.046
	R^2^	0.270		0.343		0.145		0.028	
	‘*p*’	0.040		0.043		0.042		0.015	

**Table 2 healthcare-11-02891-t002:** Multinomial regression analyses for Pittsburgh Sleep Quality Index (PSQI) components compared amongst good and poor sleep quality subjects (N = 264); *** *p* < 0.001, * *p* < 0.05.

Variables	Subjective Sleep Quality			2. Sleep Latency		
	Poor-Quality Sleep (PSQI ≥ 5) (Ref: Good-Quality Sleep (PSQI < 5))			Poor-Quality sleep (PSQI ≥ 5) (Ref: Good-Quality Sleep (PSQI < 5))		
	β	95% CI (UB)	95% CI (LB)	β	95% CI (UB)	95% CI (LB)
Actual screen time	−0.096 ***	0.541	1.525	−0.034 ***	0.548	1.706
Gender	−0.266	0.417	1.410	0.110	0.586	2.129
Age	−0.716	0.142	1.688	1.303	0.993	13.633
Parental St	−0.059	0.495	1.794	−0.405	0.336	1.323
SES	−0.113	0.779	1.024	0.164	1.022	1.358
**Variables**	**3.** **Sleep Duration**			**4.** **Sleep Efficiency**		
	**Poor-Quality Sleep (PSQI ≥ 5) (Ref: Good-Quality Sleep (PSQI < 5))**			**Poor-Quality Sleep (PSQI ≥ 5) (Ref: Good-Quality Sleep (PSQI < 5))**		
	**β**	**95% CI (UB)**	**95% CI (LB)**	**β**	**95% CI (UB)**	**95% CI (LB)**
Actual screen time	0.308 ***	0.807 ***	2.294	0.387 ***	0.878	2.469
Gender	0.717	1.105	3.793	0.634	1.030	3.450
Age	1.696	1.674	17.756	10.564	1.518	15.026
Parental St	0.090	0.539	1.861	0.075	0.582	1.998
SES	0.134	0.995	1.315	0.141	1.005	1.319
**Variables**	**5.** **Sleep Disturbances**			**6.** **Sleep Medication**		
	**Poor-Quality Sleep (PSQI ≥ 5) (Ref: Good-Quality Sleep (PSQI < 5))**			**Poor-Quality Sleep (PSQI ≥ 5) (Ref: Good-Quality Sleep (PSQI < 5))**		
	**β**	**95% CI (UB)**	**95% CI (LB)**	**β**	**95% CI (UB)**	**95% CI (LB)**
Actual screen time	1.234 ***	0.656	1.847	0.191 *	0.771	1.901
Gender	1.266	0.709	2.400	0.396	0.870	2.539
Age	3.453	0.592	7.067	10.145	1.116	8.851
Parental St	1.259	0.557	2.020	−0.018	0.576	1.675
SES	1.113	0.977	1.283	0.143	1.023	1.302
	**7.** **Daytime Dysfunction**					
	**Poor-Quality Sleep (PSQI ≥ 5) (Ref: Good-Quality Sleep (PSQI < 5))**					
	**β**		**95% CI (UB)**		**95% CI (LB)**	
Actual screen time	−0.077 ***		0.565		1.517	
Gender	0.251		0.724		2.284	
Age	0.818		0.782		6.565	
Parental St	0.067		0.600		1.905	
SES	0.102		0.975		1.258	

## Data Availability

The data presented in this study are available on request from the corresponding author. The data are not publicly available to ensure best privacy to participant data and oblige our institutional policies.

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
