# Peer review of "Effect of Smartphone Use on Sleep in Undergraduate Medical Students: A Cross-Sectional Study"

_healthcare, 2023, doi:10.3390/healthcare11212891_

Round 1

Reviewer 1 Report

Comments and Suggestions for Authors

The evaluated manuscript focuses on the relationship between smartphone use and sleep quality in a sample of Indian university students. The article's topic is relevant, and the results concerning the negative effects of excessive smartphone use on sleep could make a valuable contribution to the scientific literature. However, I believe that the manuscript has certain shortcomings that should be addressed.

Introduction:

The introduction is concise and well-structured. However, it predominantly attributes the negative relationship with sleep quality to exposure to blue light at night. While there are some studies that support this association, it's worth noting that the results in this area are controversial and could benefit from discussion. Additionally, it would be beneficial to mention other potential mediating variables in the relationship between smartphone use and poor sleep quality, such as bedtime procrastination or heightened arousal.

Methods:

Was any calculation of the sample size needed for the intended analyses performed prior to the start of the study?

The reliability of the PSQI for the sample used should be presented.

It is mentioned that the perception of smartphone overuse in terms of health hazards was assessed. How was this assessment conducted?

I recommend adding a subsection on data analysis to describe the statistical techniques used. Additionally, consider addressing whether the normality of the variables and the fulfillment of assumptions for the regression analyses have been evaluated.

In the objectives and results, bedtime and rise time are mentioned as study variables, but they do not appear in the Sleep questionnaire subsection.

Results:

Line 153: add the standard deviation of the variable age.

The methodology refers to 'perception of smartphone overuse in terms of health hazards,' while the results refer to 'perceived screen time.' Are these the same variable? If so, it would be beneficial to consistently use the same name throughout.

Throughout the manuscript, some variables are given different names, which can make the text challenging to read.

In the hierarchical regression analysis to predict Perceived Screen time, is the same variable being predicted used as a predictor in step 2?

Discussion:

Similar to the introduction, the discussion predominantly centers on the role of blue light as a mediator between smartphone use and sleep quality. It would be beneficial to explore and propose other potential mediating variables in this relationship.

Line 303 discusses smartphone addiction. While this term is commonly used in the scientific literature, it's essential to clarify that major diagnostic manuals do not include this disorder. Therefore, it might be more appropriate to refer to 'problematic smartphone use.

Author Response

Reviewer 1

Point 1: The evaluated manuscript focuses on the relationship between smartphone use and sleep quality in a sample of Indian university students. The article's topic is relevant, and the results concerning the negative effects of excessive smartphone use on sleep could make a valuable contribution to the scientific literature. However, I believe that the manuscript has certain shortcomings that should be addressed.

In the objectives and results, bedtime and rise time are mentioned as study variables, but they do not appear in the Sleep questionnaire subsection.

Reply from authors: Thank you very much for your comments. We are glad that the respected reviewer liked our work, and we sincerely thank them for their kind words of appreciation. Thank you for pointing out the discrepancy in objectives; we have now rephrased that line to give readers a better idea.

Point 2 Introduction: The introduction is concise and well-structured. However, it predominantly attributes the negative relationship with sleep quality to exposure to blue light at night. While there are some studies that support this association, it's worth noting that the results in this area are controversial and could benefit from discussion. Additionally, it would be beneficial to mention other potential mediating variables in the relationship between smartphone use and poor sleep quality, such as bedtime procrastination or heightened arousal.

Reply from authors: Thank you very much for pointing that. We have now expanded the introduction and discussion to include studies on both sides on spectrum.

Point 3: Methods: Was any calculation of the sample size needed for the intended analyses performed prior to the start of the study?

The reliability of the PSQI for the sample used should be presented.

It is mentioned that the perception of smartphone overuse in terms of health hazards was assessed. How was this assessment conducted?

I recommend adding a subsection on data analysis to describe the statistical techniques used. Additionally, consider addressing whether the normality of the variables and the fulfillment of assumptions for the regression analyses have been evaluated.

Reply from authors: Thank you very much for pointing that out.

  1. We added a line explaining sampling methodology and sample size calculation.
  2. PSQI is an established and validated scale used globally to assess sleep quality with an overall reliability coefficient (Cronbach's alpha) of 0.736. We have added a line for this as well.
  3. Further, the assessment of smartphone overuse in terms of health hazards was done using a questionnaire as described in our methods section:

The questionnaire had two parts: 1) Demographic data of age, gender, MBBS professional, socioeconomic status (SES), family structure, personal habits (smoking, alcohol consumption), use of a smartphone (hour/day), and the perception of smartphone overuse in terms of health hazards. 2) Sleep parameters measured through the Pitts-burgh Sleep Quality Index (PSQI) (overall reliability coefficient (Cronbach's alpha) - 0.736). This included questions about the participant’s usual sleep habits (mostly days and nights) during the past month. The PSQI primarily determines sleep quality using seven sleep assessment indicators: (1) subjective sleep quality, (2) sleep latent period, (3) sleep time, (4) sleep efficiency, (5) sleep difficulties, (6) daytime functional impairment, and (7) use of sleeping pills. Scores on each item range from 0 to 3 points; hence, total scores range from 0 to 21 points, with a total PSQI score of <5 indicating good sleep quality, whereas a score of ≥5 indicates poor sleep quality.

  1. We have added a paragraph on data analysis now. Thanks for the detailed help and recommendations.

Point 4: Results: Line 153: add the standard deviation of the variable age.

The methodology refers to 'perception of smartphone overuse in terms of health hazards,' while the results refer to 'perceived screen time.' Are these the same variable? If so, it would be beneficial to consistently use the same name throughout.

Throughout the manuscript, some variables are given different names, which can make the text challenging to read.

In the hierarchical regression analysis to predict Perceived Screen time, is the same variable being predicted used as a predictor in step 2?

Reply from authors: We thank the reviewer for highlighting that we have added the SD with mean.

The perception of smartphone overuse in terms of health hazards is our objective, which has been met using the results of the PSQI questionnaire. To be objective and to make our data collection reliable, we have used two measures of screentime – one as perceived by study participants and the other as reported by screentime analysis of their smartphones (as seen in the settings option). We have reviewed the manuscript again and ensured consistent usage of terms.

In the hierarchical regression analysis to predict Perceived Screen time, is the same variable being predicted used as a predictor in step 2?

Yes. ‘Perceived screen time’ is predicted and then used as a predictor in ‘Step 2 and ‘Step 3 of hierarchical regression analysis.

Point 5: Discussion: Similar to the introduction, the discussion predominantly centers on the role of blue light as a mediator between smartphone use and sleep quality. It would be beneficial to explore and propose other potential mediating variables in this relationship.

Line 303 discusses smartphone addiction. While this term is commonly used in the scientific literature, it's essential to clarify that major diagnostic manuals do not include this disorder. Therefore, it might be more appropriate to refer to 'problematic smartphone use.

Reply from authors: Thank you for highlighting the same; we have expanded on both ends of the spectrum and have been more open to the other side of the literature.

We have corrected line 303 for the term used. Many thanks for pointing out the technical discrepancy. We really appreciate it.

Reviewer 2 Report

Comments and Suggestions for Authors

Dear Authors,

You have taken a challenging topic of smartphone/light and sleep relation. It is a very complicated relation therefore it requires a profound data collection and analysis. And here there are some problems with your study. Please find below my suggestions:

1. "Supplemental file 1" is missing in the editorial system.

2. In what language was the PSQI Questionnaire (English? If so - how did you ascertain that it was fully comprehensive to all participants?)

3. How were levels of socio-economic status defined?

4. Were most/all of your participants living in their homes? There was no population living in students' houses/dormitories? Such a place of living would be a powerful factor influencing the quality of sleep

5. How exactly was time of using smartphones defined? Was it a subjective impression of the users? Or was it measurement by some application? Did you have any questions about using smartphones during the evening/night? It is using of blue light in the dar phase of the day that is most important in terms of inhibition of melatonine.

6. Presenting raw data on PSQI scores and data on usage of smartphones would be useful.

Comments on the Quality of English Language

Minor English editing would be useful.

Author Response

You have taken a challenging topic of smartphone/light and sleep relation. It is a very complicated relation therefore it requires a profound data collection and analysis. And here there are some problems with your study. Please find below my suggestions:

  1. "Supplemental file 1" is missing in the editorial system.

Reply from authors: Many thanks for pointing that out. We will provide the file again to the journal. For current reference, please use the link to the questionnaire - https://docs.google.com/forms/d/e/1FAIpQLSdhYOCAMb1flx154EiGGKW7PO7x48FU-j1SIG4LOX1nyq3ckg/viewform

Point 2: In what language was the PSQI Questionnaire (English? If so - how did you ascertain that it was fully comprehensive to all participants?)

Reply from authors: Thank you very much for your comments. The questionnaire was used in English. Since all students could read and understand English, usage was never a problem. The PSQI is already validated in the English language, so reliability was never a problem.

Point 3: How were levels of socio-economic status defined?

Reply from authors: Thank you very much for your comments. The socioeconomic status was determined as per the kuppuswamy socioeconomic scales. The scale is the most common scale used in the region and gives the best representation. Moreover, our sample population of MBBS students was already aware of the scale used, so their judgment was reliable.

Point 4: Were most/all of your participants living in their homes? There was no population living in students' houses/dormitories? Such a place of living would be a powerful factor influencing the quality of sleep

Reply from authors: Thank you very much for your comments. The study participants were hostelers, being MBBS students, presenting a common ground. But indeed, the point raised by you is important and forms a limitation to generalizability, and hence, we have expanded on the limitations section:

“Since all students were hostelers, the study cannot be generalized to those students living at home/day scholars – which can add potential unknown confounders”

Point 5: How exactly was time of using smartphones defined? Was it a subjective impression of the users? Or was it measurement by some application? Did you have any questions about using smartphones during the evening/night? It is using of blue light in the dar phase of the day that is most important in terms of inhibition of melatonine.

Reply from authors: Thank you very much for your comments.

To be objective and to make our data collection reliable, we have used two measures of screen time – one as perceived by study participants and the other as reported by screen time analysis of their smartphones (as seen in the settings option).

Point 6: Presenting raw data on PSQI scores and data on usage of smartphones would be useful.

Further, as pointed out by the reviewers, we’ve thoroughly checked and revised the manuscript for any possible English language errors.

Reply from authors: Thank you very much for your comments. We are open to sharing the data as and when asked by the editor/reviewer/ anyone upon request with the corresponding author.

We sincerely thank the respected editor and all the respected reviewers for their help in improving our article.

Round 2

Reviewer 1 Report

Comments and Suggestions for Authors

The authors have addressed all the comments and questions well. 

One more question for approval: In the sample size calculation, what effect size was it calculated for?

Author Response

The authors have addressed all the comments and questions well. 

One more question for approval: In the sample size calculation, what effect size was it calculated for?

Reply from authors: Thank you very much for your comments. We are glad that the respected reviewer liked the revisions. We used an effect size of 0.036 – per the statistical tests planned (as shown in image of G Power software attached).

We have added this detail to manuscript as well now. Please kindly see the figure in the attached pdf. Thank you very much. 
